# Dermatology-Related Emergency Department Visits in Tertiary Care Center in Riyadh, Saudi Arabia: A Descriptive Study

**DOI:** 10.3390/healthcare12232332

**Published:** 2024-11-22

**Authors:** Abdullah Alshibani, Saif Osama Alagha, Abdulmohsen Jameel Alshammari, Khaled Jameel Alshammari, Abdulelah Saeed Alghamdi, Khalid Nabil Nagshabandi

**Affiliations:** 1Emergency Medical Services Department, College of Applied Medical Sciences, King Saud bin Abdulaziz University for Health Sciences, Riyadh 11481, Saudi Arabia; alshammarik148@gmail.com; 2King Abdullah International Medical Research Center, Riyadh 11481, Saudi Arabia; saifalagha01@gmail.com (S.O.A.); abdulmohsens.2001@gmail.com (A.J.A.); 3College of Medicine, King Saud bin Abdulaziz University for Health Sciences, Riyadh 11481, Saudi Arabia; 4Department of Dermatology, College of Medicine, King Saud bin Abdulaziz University for Health Sciences, Riyadh 11481, Saudi Arabia; alghamdi.ast@gmail.com; 5Department of Dermatology, College of Medicine, King Saud University, Riyadh 11481, Saudi Arabia; khaloed23@gmail.com

**Keywords:** skin, drug reaction, vasculitis, subcutaneous, inflammatory, infection, urticaria, angio-edema, urgent, ambulatory

## Abstract

**Background/Objectives**: Dermatological complaints are commonly seen in the emergency department (ED) setting and may be attributed to infectious, inflammatory, allergic, hypersensitivity, or traumatic processes, yet few studies have been carried out in Saudi Arabia addressing this topic. This study, therefore, aimed to explore this issue by investigating the most common dermatology-related ED encounters in a large tertiary care center in Riyadh, Saudi Arabia, and estimating the incidence of these encounters. **Methods**: This was a retrospective cohort study conducted in the ED of King Abdulaziz Medical City, a tertiary care center in Riyadh, Saudi Arabia. Data included all patients with dermatology-related ED visits during the period of 2022–2023. Demographic information including, for example, age and sex was collected. The International Classification of Diseases, 10th Revision (ICD-10) was used for the classification of diagnoses. **Results**: A total of 11,443 patients were included in the study, with male patients making up the majority (54.9%). The mean age upon diagnosis was 22.4 ± 23.2 years. More than half of the patients (55.3%) were diagnosed during childhood (<18), while proportions of older ages declined gradually. Average monthly presentations ranged from 400 to 560. Rash and non-specific skin eruptions (16%), cellulitis (13.6%), and urticaria (12.2%) were the most frequent dermatological emergencies. **Conclusions**: This study examined the dermatological conditions commonly seen in the emergency department. The findings highlighted a range of dermatology diseases that are typically seen in the ED. Addressing these prevalent disorders in the future will enhance ER physicians’ understanding and management of such common dermatological problems.

## 1. Introduction

Dermatological emergencies involve various conditions such as infectious skin diseases, adverse drug reactions, vasculitis, flares of inflammatory skin diseases, urticaria, and angio-edema [1]. Dermatological conditions account for about 4–8% of all emergency department (ED) visits; however, only some of such visits are considered to be a true emergency [2]. Most dermatological diseases are known to be not life-threatening and are mostly managed in an outpatient setting; however, dermatological emergencies may end up resulting in high mortality and morbidity. Furthermore, skin changes might be the only sign and indication of a serious systemic disease. Therefore, early recognition with appropriate investigations and correct interventions is essential in preventing complications and consequences of such presentations [3,4,5].

A cohort cross-sectional study done in the United States of America (USA) revealed that over a period of 10 years, an estimated 51,809,000 visits had a primary diagnosis of a dermatological disease, accounting for 3.8% of all (ED) visits, with cellulitis and cutaneous abscess (1.2–1.3%) being the most common diagnoses in 2016 and 2017 [6]. Moreover, a study done in Los Angeles, USA showed that out of 204 patients, the most frequent diagnoses among the non-admitted patients were eczematous dermatitis not otherwise specified (8.9%), scabies (7.2%), contact dermatitis (6.6%), and cutaneous drug eruption (6.0%) [7]. While of those admitted, the most prevalent conditions were erythema multiforme major/Stevens–Johnson syndrome (22%) and pemphigus vulgaris (14%) [7]. A prospective study done in Spain showed that there were 3084 visits due to a dermatology-related complaint, comprising 5.6% of (ED) visits over the 12-month study duration, and the most frequent diagnoses were acute urticaria (7.6%), contact dermatitis (6.1%), drug-induced reactions (4.6%), cellulitis (3.8%), and herpes zoster (3.7%) [8]. In India, a study conducted over a one-year period showed that the total cases of dermatological conditions were 327 accounting for 0.92% of (ED) visits [9]. Additionally, the most frequently encountered conditions in the out-patient cases were urticaria and angioedema (29.7%), viral exanthem (25.9%), and drug rashes (15.9%) [9].

In Saudi Arabia, a study done in Qassim region in 2011 reported a total of 1147 skin cases accounting for 0.8% of whole ED visits. The study reported that dermatitis and eczematous disorders (48.8%) were the most common disorders, followed by viral infections (21.2%) and urticaria (10.5%), respectively [10].

Although there is some evidence about dermatological emergencies in Saudi Arabia, the number of studies covering this topic is low. Moreover, these published studies included a low sample size and/or included patients over a short period of time. This, in turn, highlights the need for additional studies including a larger population and over a long period for better characterization and representation of patients with dermatological emergencies in Saudi Arabia. Therefore, this study aimed to address this issue by describing the patients with dermatological disorders who required ED visits, and investigating the most common encounters of dermatology-related ED visits at King Abdulaziz Medical City (KAMC) in Riyadh from January 2022 until December 2023. 

## 2. Materials and Methods

### 2.1. Study Design

This study was a retrospective cohort study based on a chart review of the patients admitted to the emergency department between January 2022 and December 2023. It was an observational study and the primary source of information was patient medical charts using the “BestCare” system, which is an electronic patient medical records system that is employed at KAMC. Ethical approval was obtained from the institutional review board at King Abdullah International Medical Research Center, Riyadh, Saudi Arabia (Study Number: NRC24R/107/02).

### 2.2. Study Setting

The study was conducted in the ED at KAMC, Riyadh, which is considered one of the most comprehensive healthcare medical cities in Saudi Arabia. It was established in 1982, currently has 1973 operational beds, and employs about eight thousand health and medical support professionals.

### 2.3. Study Population

All patients who presented to the emergency department with dermatology-related conditions were included in this study. Patients with missing demographic information such as age or sex were excluded from the study, as well as those with missing dermatology-related emergency information like the nature of the dermatology emergency. A total of 11,443 patients were included in the study.

### 2.4. Data Collection and Management

Data were collected by the research team members using the files of patients who presented to the emergency department between January 2022 and December 2023 at KAMC. These files were analyzed and reviewed using a data collection sheet. The data collection sheet contained the patients’ demographic data including age, sex, country of origin, and comorbidities. It also included dermatology-related emergency information (date of ED presentation and main diagnosis based on the ICD-10 codes). All collected data were anonymized and kept secured at a university-secured desktop where only the study team members had access to this data.

### 2.5. Data Analysis

Data analysis was performed using RStudio (R version 4.3.1). We expressed variables as frequencies and percentages. A line chart was developed to present monthly frequencies of dermatology emergency cases. The most common age, by year, upon diagnosis was visualized in a histogram, and age categories were created and expressed as younger than 18, from 18 to 29 years, 30 to 44 years, 45 to 59 years, and 60 years or older. The International Classification of Diseases, 10th Revision (ICD-10) was used for classification of diagnoses.

## 3. Results

This study included records of 11,443 patients. This represented 2.30% of total ED visits from Jnaury 2022 to December 2023 (497,424 ED visits). The mean age upon diagnosis was 22.4 ± 23.2 years, the oldest patient was 103 years old, and the youngest was 1 month old. Male sex was higher (54.9%) than female sex (45.1%). Of the study population, 2559 (22.4%) had comorbidities, with diabetes (9.4%) being the most prevalent comorbidity, followed by dyslipidemia (6.4%) and hypertension (5.0%). The majority of patients were Saudi citizens (96.4%), and only 412 (3.6%) patients were of different nationalities. Filipino was the most common non-Saudi nationality comprising 30.1% of the non-Saudis, followed by Syrian (25.7%) and Malaysian (8.7%) (Table 1).

In terms of the age distribution at the time of diagnosis, the proportions of dermatological ED visits decreased with advancing age, with more than half of the patients (55.3%) being diagnosed in childhood (<18), decreasing to 12.5% between 18 to 29 years, 13.9% between 30 to 44 years, 8.8% between 45 to 59 years, and 9.5% for 60 years or more (Figure 1).

The number of dermatology-related emergency department visits varied monthly during 2022 and 2023. In the first 7 months of 2022 there were relatively low numbers of ED visits, ranging between 400 and 480 visits per month. However, visits increased in the following two months, exceeding 550 visits in August and September. Subsequently, the number of visits decreased to 488 and 423 in October and November 2022. ED dermatological visits generally remained above the threshold of 470 visits in the following months until August 2023. After that, the number of visits fluctuated between 400 and 480 visits per month (Figure 2).

The most frequent presentations of dermatological emergencies were for the infections of the skin and subcutaneous tissue category, including cellulitis (n = 1558, 13.6%); cutaneous abscess, furuncle, and carbuncle (n = 1001, 8.7%); and acute lymphadenitis (n = 294, 2.6%). These were followed by symptoms and signs involving the skin and subcutaneous tissue classification, which included rash and other non-specific skin eruptions (n = 1828, 16.0%). The third-most-frequent category was urticaria and erythema, with urticaria being the most diagnosed in this category (n = 1400, 12.2%). This was followed by burns and corrosions, which incorporated burn of unspecified region (n = 735, 6.4%), of ankle and foot (n = 137, 1.2%), of wrist and hand (n = 132, 1.2%), and of trunk (n = 122, 1.1%) (Table 2).

## 4. Discussion

This study described dermatology-related emergencies for patients attending the ED at a tertiary care center in Riyadh, Saudi Arabia. More than 11,000 were included in this study which, to our knowledge, to date, is the largest sample size representing the demographics and clinical characteristics of dermatology-related emergencies in Saudi Arabia. The mean age of the participants was 22.4 ± 23.2 years with males being the more likely patients to present with dermatology-related emergencies (54.9%). More than 22% of the study population had comorbidities, with diabetes (9.4%), dyslipidemia (6.4%), and hypertension (5%) being the most common. The majority of the patients were Saudi citizens (96.4%). With regard to age distribution upon diagnosis, the frequencies and proportions of dermatology-related emergencies requiring an ED visit decreased with advancing age, with more than 55% of the patients being diagnosed at the age group <18 years. This could have been due to different reasons, including that elderly patients often face other serious health conditions, such as diabetes mellitus and cardiovascular diseases, which may lead them to prioritize such comorbidities. Also, many elderly patients have familiarity with skin problems from previous encounters, which may lead them to adopt self-management strategies and prevent the emergencies for developing. The average monthly presentations of dermatology-related emergencies at the ED during the study period ranged from 400 to 560. Using the ICD-10 codes, the most common diagnoses of dermatology-related emergencies requiring an ED visit were rash and non-specific skin eruption (16%), cellulitis (13.6%), and urticaria (12.2%).

The findings of our study showed that dermatology-related ED visits represented 2.30% of total ED visits, which is similar to the findings of other international studies. For example, a study in Spain showed that dermatology-related ED visits represented 2.59% of all ED visits [11]. Another study in the United States of America showed that 3.3% of all ED visits were related to dermatology diseases [12].

While most dermatology-related visits in emergency department settings are typically not acute or life-threatening, “true dermatological emergencies” such as toxic epidermal necrolysis (TEN) and Stevens–Johnson syndrome (SJS), Rocky Mountain spotted fever, and necrotizing fasciitis can result in significant morbidity and mortality [12]. However, several previous studies have indicated that many patients presenting at the EDs with dermatological issues do not have genuine or fatal dermatological emergencies [3]. Thus, there is a need to conduct awareness campaigns about dermatological conditions to educate the community about what requires a visit to the ED and what could be handled in outpatient clinics. Also, the provision of alternative care pathways for those non-urgent cases, including outpatient dermatology, and offering the services needed remotely via teledermatology, would benefit both the patients, by saving their time and effort, and the hospital, by reducing the burden on the ED and the waste of resources.

The results from our study, involving 11,443 patients with dermatological emergencies, provide an insightful comparison with existing literature, particularly those studies conducted in Saudi Arabia. In our study, the mean age of patients presenting with dermatological emergencies was 23.4 years, with a slightly higher proportion of males (54.9%) compared to females (45.1%). The gender distribution aligned with the findings of Bahamdan et al., who reported a male-to-female ratio of (1.2:1) in their study conducted at Asir Central Hospital in Abha, southern Saudi Arabia [13]. This distribution might be attributed to males being more commonly afflicted with infectious skin diseases, which was the most common category of diagnoses in this study [14]. Also, KAMC provides healthcare services for military soldiers and trainees, who are more exposed to harsh environments and UV light, which both can be risk factors for various dermatological diseases and/or manifestations. The predominant age group in our study was consistent with younger populations frequently seeking dermatological consultations, reflecting a similar trend observed in the Bahamdan et al. study [13]. The most common dermatological diagnoses in our study were rash and other non-specific skin eruptions (16.0%), cellulitis (13.6%), and urticaria (12.2%). This distribution slightly aligns with the findings from Özkur et al., who evaluated dermatology consultations at an ED in Turkey over a one-year period. They reported that the majority of consultations were for infectious conditions (86.9%), with viral infections being the most common [3]. Inflammatory dermatoses and urticaria/angioedema followed in frequency [3]. The study also noted that 24.7% of the cases were considered “true dermatological emergencies”, which is a substantial proportion. The demographic distribution in their study, with a mean patient age of 44.6 years and a slightly higher male predominance, is somewhat different from our study, which had a younger average age of 23.4 years [3]. Bahamdan et al. also observed a high prevalence of eczema/dermatitis (25.68%) and viral infections (10.12%) in the Asir region, Saudi Arabia [13]. Another study from Turkey by Kilic D et al., who conducted a retrospective observation at a university hospital in Antalya, Turkey, included patients over 18 years old presenting with dermatological complaints. They found that the most common complaint was erythematous skin rash with pruritus (50.9%), and the most frequent final diagnosis was urticaria and drug eruptions (84.5%). Most patients (98.8%) were classified as urgent but not emergent, and only 0.2% were considered emergent. Consultations were needed for 6.4% of patients, with a hospitalization rate of 2.2%. The study found that most dermatological complaints in the ED were non-urgent and could be treated at an outpatient clinic [1].

Comparative studies from other regions also provide further context for our findings. For example, a study by Gupta et al. in North India highlighted the predominance of bacterial infections in dermatological emergencies, which aligns with the high prevalence of cellulitis observed in our study [15]. Additionally, a study by Moon et al. from the United States of America assessed pediatric dermatology consultations in the ED and found skin infections to be the most frequent cause of consultations [16]. Lastly, Abedini R et al. evaluated patients visiting a dermatology emergency unit in a university dermatology hospital in Tehran, Iran, and found that the most prevalent conditions encountered were infections and infestations (41.9%), urticaria (16.7%), and dermatitis (13.2%) [4]. A small percentage (1%) of patients were referred by another physician, with psoriasis being the most frequent diagnosis [4]. Hospitalization was required for 2.6% of the patients, with a higher rate observed among those referred by another physician [4]. Our study also revealed monthly variations in dermatology-related emergency department visits, with a noticeable increase in visits during August and September. This trend could be influenced by seasonal factors, such as increased outdoor activities and exposure to environmental triggers. Özkur et al. also noted a peak in dermatological consultations in April, suggesting that seasonal variations play a significant role in dermatological emergencies [3].

Current and previous research findings indicate that the majority of skin-related visits to the ED are not “true” cutaneous emergencies. There is a highlighted need for improved knowledge among general practitioners and emergency physicians regarding common and acute dermatological conditions [1,3]. A study conducted at King Abdulaziz University Hospital in Jeddah, Saudi Arabia, focused on assessing the ability of non-dermatologist physicians to recognize urgent skin diseases [17]. The research revealed that non-dermatologists correctly diagnosed urgent dermatological conditions 61.33% of the time based on typical presentations [18]. However, confidence in these diagnoses was notably lower, with only 25.3% expressing full confidence in their answers [18]. Herpes zoster was the most recognizable condition, while pemphigus vulgaris was the least [18]. The findings from the available literature emphasize the significant need for enhanced dermatological training among non-specialist physicians to improve diagnostic accuracy and patient outcomes.

Drug eruptions can lead to severe cutaneous manifestations, such as Stevens–Johnson syndrome or toxic epidermal necrolysis, which may often present in the ED due to their acute and potentially life-threatening nature. A study conducted at KAMC in Riyadh, Saudi Arabia, provides a comprehensive evaluation of severe cutaneous adverse drug reactions (SCARs) over a five-year period [19]. Out of 3050 dermatology consultations, 253 cases were identified as cutaneous adverse drug reactions, with 41 cases classified as SCARs, constituting 16.2% of the adverse reactions [19]. The research highlighted that antibiotics and anticonvulsants were the most common causative agents, accounting for 68.3% and 22% of the cases, respectively [19]. DRESS was the predominant SCAR, with vancomycin being the most frequently implicated drug [19]. SJS/TEN had the highest mortality rate at 45.5% [19]. These findings underscore the rarity but significant impact of SCARs in the region [19].

The consistent findings from the previous studies of high infection rates, particularly viral and bacterial infections, are all consistent with the high incidence of cellulitis in our study, underscoring the importance of effective infection control measures in emergency care settings to prevent further superinfection that might lead to higher morbidity and mortality rates. The lower incidence of pyogenic infections in our study, as well as in another study from Saudi Arabia [13] compared to regions like India, suggests that socioeconomic factors and healthcare infrastructure significantly influence the epidemiology of dermatological conditions. The findings from this study highlight several potential strategies to improve the management of dermatology-related emergency department visits, particularly through better triage and tailored care pathways. Given the frequency of non-urgent dermatological cases, implementing structured outpatient and teledermatology pathways may help reduce ED congestion and improve patient triage, ensuring critical cases receive timely attention. Additionally, educating primary care and ED practitioners about common dermatological emergencies could enhance early diagnosis and decision-making, particularly in differentiating between urgent and non-urgent presentations. These steps could improve patient outcomes, optimize resource allocation, and potentially reduce overall healthcare costs by diverting manageable cases to outpatient settings.

This study has several implications. It is the largest study, to our knowledge, describing dermatology-related ED visits in Saudi Arabia. The findings of this study could help healthcare professionals working in the emergency care settings to understand the demographics and clinical characteristics of patients attending their ED with dermatology-related complaints. It also helps current and future research to understand the nature of dermatology-related ED visits and apply interventions to address the most common presentations, especially those with high mortality and morbidity rates. Moreover, our findings underscore the potential value of establishing multicenter studies to broaden the scope of dermatological emergency research across varied demographics and healthcare settings. Such collaborative research could refine our understanding of dermatology-related emergency department visits, support the identification of region-specific trends, and enhance emergency response protocols. Additionally, prospective studies with comprehensive demographic data, including ethnicity, could provide deeper insights into genetic and environmental influences on dermatological conditions, further tailoring patient care and resource allocation. However, some evident limitations need to be highlighted. First, data were obtained from a single center which may have impacted the generalizability of the study findings. Furthermore, the retrospective nature of the study may have introduced some inherent limitations as it relied on previously recorded data which could have been incomplete, inconsistent, or have include important variables that could have been valuable for this study (including, for example, ethicity and true dermatoligical emergencies). In addition, a key limitation of our study was that we could not analyze the links between different age groups and dermatological diseases. The dataset lacked sufficient age stratification and did not include enough individuals across various age ranges for us to draw meaningful conclusions. Moreover, comparing a wide range of dermatological conditions in depth would require extensive research and data collection, which was not possible within the scope and timeframe of this study. Future research should aim to fill these gaps to improve our understanding of how dermatological conditions vary with age. For diagnosis and treatment of dermatological cases in the ED, our data lacked information about who was making the diagnosis and determining the treatment—the ED physician, dermatology consultant, or both. Therefore, we were unable to calculate how many cases were diagnosed/treated by an ED physician, dermatology consultant, or both. However, from our understanding of the study setting, it is the responsibility of the ED physician to make the diagnosis and initiate the treatment in the ED. For the ED physician to make a diagnosis, dermatology consultations are requested in complex and ambiguous cases or if the patient needs to be admitted in the hospital. Lastly, the study may have introduced a risk for selection bias as only patients with completely documented and stored documents during the study period were included.

## 5. Conclusions

In conclusion, the study underscores the importance of recognizing and managing common dermatological emergencies in the ED setting, where infections, inflammatory conditions, and trauma are frequently encountered. Addressing these prevalent issues through improved educational programs for healthcare providers and establishing efficient outpatient or teledermatology pathways can help optimize patient outcomes and alleviate ED burden. Expanding similar research across the region will enhance our understanding and management of dermatological emergencies, ultimately benefiting patient care and resource allocation.

## Figures and Tables

**Figure 1 healthcare-12-02332-f001:**
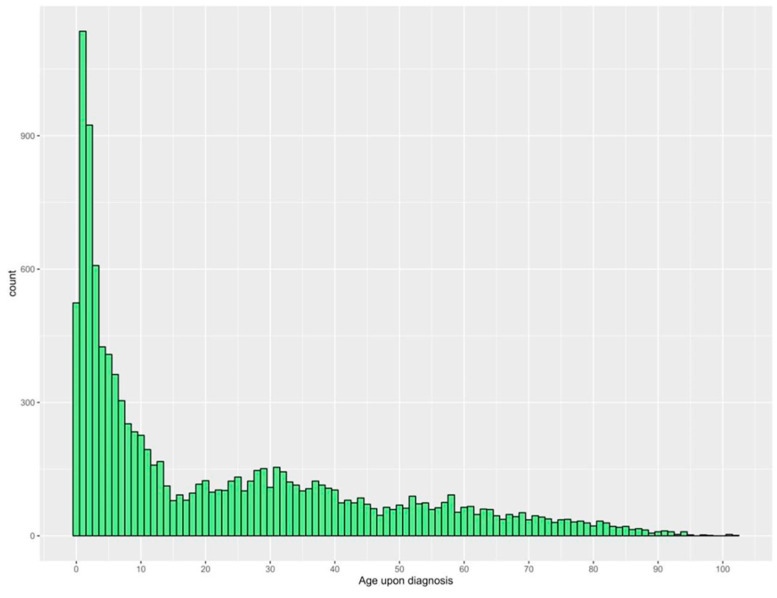
The distribution of patients’ age upon the diagnosis of dermatological emergencies (years).

**Figure 2 healthcare-12-02332-f002:**
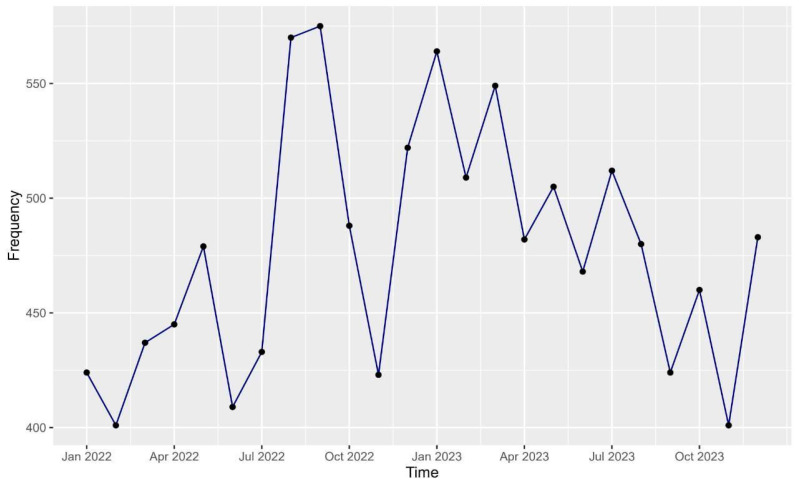
The number of dermatology-related emergency department visits by month.

**Table 1 healthcare-12-02332-t001:** Characteristics of patients.

Characteristic	Description
Age at diagnosis (years)	22.4 ± 23.2
Gender	
Male	6285 (54.9%)
Female	5158 (45.1%)
Comorbidities	
Diabetes	1077 (9.4%)
Dyslipidemia	736 (6.4%)
Hypertension	576 (5.0%)
Stroke	122 (1.1%)
Heart failure	48 (0.4%)
Nationality	
Saudi	11,031 (96.4%)
Non-Saudi	412 (3.6%)
Non-Saudi nationalities	
Filipino	124 (30.1%)
Syrian	106 (25.7%)
Malaysian	36 (8.7%)
Other Asian nationalities countries	74 (18.0%)
Other African nationalities	48 (11.7%)
Other European nationalities	11 (2.7%)
USA	10 (2.4%)
Canada	2 (0.5%)
Australia	1 (0.2%)

Mean ± SD; n (%).

**Table 2 healthcare-12-02332-t002:** The distribution of patients diagnoses.

Characteristic	Description n (%)
**Infections of the skin and subcutaneous tissue**	**3529 (30.8%)**
Cellulitis	1558 (13.6%)
Cutaneous abscess, furuncle, and carbuncle	1001 (8.7%)
Acute lymphadenitis	294 (2.6%)
Pilonidal cyst	258 (2.3%)
Other local infections of skin and subcutaneous tissue	241 (2.1%)
Impetigo	176 (1.5%)
Staphylococcal scalded skin syndrome	1 (0.0%)
**Symptoms and signs involving the skin and subcutaneous tissue**	**1850 (16.2%)**
Rash and other non-specific skin eruption	1828 (16.0%)
Other skin changes	22 (0.2%)
**Urticaria and erythema**	**1430 (12.5%)**
Urticaria	1400 (12.2%)
Other erythematous conditions	16 (0.1%)
Erythema multiforme	12 (0.1%)
Erythema nodosum	1 (0.0%)
Erythema in diseases classified elsewhere	1 (0.0%)
**Burns and corrosions**	**1353 (11.8%)**
Burn and corrosion, body region unspecified	735 (6.4%)
Burn and corrosion of ankle and foot	137 (1.2%)
Burn and corrosion of wrist and hand	132 (1.2%)
Burn and corrosion of trunk	122 (1.1%)
Burns and corrosions of multiple body regions	69 (0.6%)
Burn and corrosion of hip and lower limb, except ankle and foot	52 (0.5%)
Burn and corrosion of head and neck	47 (0.4%)
Burn and corrosion of shoulder and upper limb, except wrist and hand	45 (0.4%)
Burns classified according to extent of body surface involved	14 (0.1%)
**Viral infections characterized by skin and mucous membrane lesions**	**1322 (11.6%)**
Other viral infections characterized by skin and mucous membrane lesions, not classified elsewhere	459 (4.0%)
Herpesviral [herpes simplex] infections	397 (3.5%)
Unspecified viral infection characterized by skin and mucous membrane lesions	140 (1.2%)
Zoster [herpes zoster]	112 (1.0%)
Varicella [chickenpox]	84 (0.7%)
Measles	74 (0.6%)
Viral warts	48 (0.4%)
Monkeypox	8 (0.1%)
**Other disorders of the skin and subcutaneous tissue**	**780 (6.8%)**
Ulcer of lower limb, not classified elsewhere	179 (1.6%)
Granulomatous disorders of skin and subcutaneous tissue	156 (1.4%)
Decubitus ulcer and pressure area	148 (1.3%)
Other disorders of skin and subcutaneous tissue, not classified elsewhere	137 (1.2%)
Atrophic disorders of skin	81 (0.7%)
Vasculitis limited to skin, not classified elsewhere	25 (0.2%)
Other disorders of pigmentation	10 (0.1%)
Hypertrophic disorders of skin	10 (0.1%)
Corns and callosities	9 (0.1%)
Lupus erythematosus	6 (0.1%)
Pyoderma gangrenosum	4 (0.0%)
Other epidermal thickening	3 (0.0%)
Other localized connective tissue disorders	3 (0.0%)
Seborrhoeic keratosis	3 (0.0%)
Other disorders of skin and subcutaneous tissue in diseases classified elsewhere	3 (0.0%)
Vitiligo	2 (0.0%)
Acanthosis nigricans	1 (0.0%)
**Dermatitis and eczema**	**680 (5.9%)**
Atopic dermatitis	504 (4.4%)
Other dermatitis	176 (1.5%)
**Disorders of skin appendages**	**318 (2.8%)**
Nail disorders	140 (1.2%)
Follicular cysts of skin and subcutaneous tissue	82 (0.7%)
Other follicular disorders	50 (0.4%)
Acne	22 (0.2%)
Nail disorders in diseases classified elsewhere	10 (0.1%)
Alopecia areata	4 (0.0%)
Other non-scarring hair loss	4 (0.0%)
Eccrine sweat disorders	3 (0.0%)
Hair colour and hair shaft abnormalities	1 (0.0%)
Cicatricial alopecia [scarring hair loss]	1 (0.0%)
Rosacea	1 (0.0%)
**Neoplasms**	**94 (0.8%)**
Benign lipomatous neoplasm	65 (0.6%)
Other malignant neoplasms of skin	14 (0.1%)
Malignant melanoma of skin	8 (0.1%)
Melanoma in situ	4 (0.0%)
Melanocytic naevi	2 (0.0%)
Other benign neoplasms of skin	1 (0.0%)
**Papulosquamous disorders**	**53 (0.5%)**
Psoriasis	24 (0.2%)
Pityriasis rosea	19 (0.2%)
Other papulosquamous disorders	7 (0.1%)
Lichen planus	2 (0.0%)
Parapsoriasis	1 (0.0%)
**Other congenital malformations**	**19 (0.2%)**
Epidermolysis bullosa	9 (0.1%)
Other congenital malformations of skin	7 (0.1%)
Congenital ichthyosis	3 (0.0%)
**Frostbite**	**8 (0.1%)**
Superficial frostbite	6 (0.1%)
Frostbite involving multiple body regions and unspecified frostbite	2 (0.0%)
**Radiation-related disorders of the skin and subcutaneous tissue**	**3 (0.0%)**
Sunburn	3 (0.0%)
**Bullous disorders**	**2 (0.0%)**
Other bullous disorders	1 (0.0%)
Bullous disorders in diseases classified elsewhere	1 (0.0%)
**Mycoses**	**2 (0.0%)**
Chromomycosis and phaeomycotic abscess	2 (0.0%)

## Data Availability

The raw data supporting the conclusions of this article will be made available by the principal investigator upon request.

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
