# Peer review of "Dermatology-Related Emergency Department Visits in Tertiary Care Center in Riyadh, Saudi Arabia: A Descriptive Study"

_healthcare, 2024, doi:10.3390/healthcare12232332_

Round 1

Reviewer 1 Report

Comments and Suggestions for Authors

Dear Authors,

It is gratifying that this topic is addressed for the first time in Saudi Arabia.

This manuscript aimed to investigate this issue by estimating and investigating the frequency of the most common dermatology-related emergency department encounters in a large tertiary care center in Riyadh, Saudi Arabia. The data included patients with dermatology-related emergency department visits during the period 2022-2023. Demographic information such as age and gender was collected. The classification of diagnoses was made according to the International Classification of Diseases.

In the current study, the authors only summarized a few aspects of the topic to some extent, and there is a lack of effective and accessible information. Improvement of the manuscript in terms of innovative information and evaluation would add value to the study.

major revision

Insufficient details were provided in terms of statistics.

For example,

….Data were analyzed using r sudio ….package Version ….. (R programme). Qualitative variables were compared using the chi-square test or Fisher exact test when necessary, that is, when the value in any of the cells in the contingency table was lower than 5. Statistical significance was set at P < .05.

Figure 2 (Monthly distribution) is appreciated.

However, the originality of the manuscript could have been strengthened by analyzing it as summer or winter months. Our study also reveals that there are 223 monthly changes in dermatology-related emergency department visits and a significant increase in visits in August and September. This could be related to the regional location and geographical situation.

The discussion is generally positive.

Reviewer 2 Report

Comments and Suggestions for Authors

The authors have presented the article nicely. However, some corrections may improve the quality of the article:

1. The authors have mentioned some basic information regarding dermatological emergency. They did not mention any association or correlation between age group and various diseases. They can also make a comparison between different dermatological disease and could try to find out the cause of the increasing or decreasing the disease.

2. Caption of the Table 2 needs to be corrected.

Comments on the Quality of English Language

English language can be improved. They can write simple sentence rather than complex sentence. 

Reviewer 3 Report

Comments and Suggestions for Authors

This a well-written paper, but of limited interest, as : 1) data were drawn from a single center and 2) there is just demographic data description with minimal critical analysis. TThe discussion section, typically reserved for in-depth analysis, is used here mainly to repeat the results section findings.

Some key considerations for improving the quality of the paper are the following:

1. While you accurately note that most cases presenting to the dermatology ER are neither urgent nor emergent, the paper misses an opportunity to assess whether ER resources are being used appropriately. It would benefit from evaluating whether alternative care pathways, such as outpatient dermatology or teledermatology, might be more suitable (or already followed in your department) for managing these non-urgent cases.

2. The paper could have provided valuable insights into the broader healthcare implications, such as potential over-reliance on emergency services for non-emergent dermatologic issues, which can burden ER units. This might suggest a need for better patient education on what constitutes a dermatologic emergency and underscore the importance of accessible dermatologic services in outpatient settings.

3. It is mentioned that dermatology-related emergencies requiring ED visits decrease with advancing age (lines 170-171), despite the potential for serious dermatologic conditions in the elderly. Consider expanding on this point and explain the reasons (e.g., elderly patients may prioritize co-morbid conditions over dermatologic issues, or they may have greater access to regular healthcare services?)

4. Similarly, consider analyzing the fact that men seem to visit more frequently the ER in your study

Reviewer 4 Report

Comments and Suggestions for Authors

I reviewed the manuscript with the title “Dermatology-related Emergency Department Visits in Tertiary 2 Care Center in Riyadh, Saudi Arabia: A Descriptive Study”.

Abstract of the study is well written.

Introduction is well written with references to similar studies in other countries.

Materials and methods: for study population duration of the study is mentioned with data collected from retrospective study. Number of patients included in the study can be added here.

Data collection and management: authors have mentioned collecting demographics data including age, gender, country of origin and comorbidities. Was ethnicity of the patients captured? Would it be possible for this study to capture or mention if done so already as genetics play an important role in dermatological inflammations?

Can mention age range included in the study (youngest to oldest patient)

Results:

Line 122- please add percentage for female patients.

Line 124-126- nationality of patients with percentage is mentioned. Would be useful to how nationality matters for study outcome. Ethnicity would be more meaningful.

Line 159: provide table heading properly. At the moment it’s the journal guidelines.

Discussion: line 166 states more male patients were observed in the ED clinic. Is there any specific reason that observed dermatological inflammations are more prevalent in men than women? References form other studies can be provided here if any.

Line 177-178: authors state that although most ED cases are not life threatening but some can be. it is not mentioned how many of such life threatening cases were observed in the current study cohort.

Line 193: this distribution slightly aligns with….- please rephrase the sentence replacing the word “slightly aligns”.

Line 198: author may define what are “true dermatological emergencies” in terms of classification as it has been used a few times in this manuscript.

authors may add how the findings of this study will help dermatology clinics going forward. Will/can the current practice be refined based on the findings. what will b the next stage for this study? for example expand the study making it multi centre, plan prospective study etc.

Overall this is the largest study of the dermatological emergencies in clinic that’s studied so far. Ethnicity can be included that will provide information on if patients genetic background plays a role. Classification for “true dermatological emergencies” can be provided.

Conclusion needs to be re-written concluding the study without describing the study again.

Reviewer 5 Report

Comments and Suggestions for Authors

Dear Authors,

Thank you very much for the study on a topic that is certainly underestimated - dermatological cases in the emergency department, a considerable proportion of which are certainly not life-threatening and acute emergencies and thus often represent an annoying burden on emergency departments. 

The topic was probably worth researching systematically, so I think the work is worth publishing.

Best regards and wishes!

My comments below:

- line 47: Please name the most important of these dermatological emergencies with high risk (+ references) [acc. line 176 ff]

- line 85: please clarify, e.g., 01/2022 - 12/2023 (2022 + 2023?) 

- line 131: age at time of diagnosis - please add, e.g.: diagnosis "during emergency visit" (not pre-existing conditions)

-line 138: monthly distribution - for 2 years a statistical analysis is not meaningful - but what about a small insert/table with the sums for every month (Jan 2022 + Jan 2023) - to help your readers to recognise your discussion in line 224 (please check)? Maybe you should write to the axis every month (Fig. 2)?

- line 180 "early diagnosis and appropriate treatment of these emergencies are crucial" should appear as "take home message" again in conlusions (Don´t overlook real emergencies in the midst of dermatological conditions that may have time!)

- discussion: very long, try to shorten / try to use subheadings

- abstract: last sentence promises insights in "management" of dermatological problems ... Please take this up again in the discussion and explain it in more detail.

- Figures: check the size of the axis labels, please make them larger and more uniform

- Table 1/Figure 1: Ok, it's obvious - but maybe add unit for age: "[years]"?

Round 2

Reviewer 1 Report

Comments and Suggestions for Authors

Thank you for the revision

Comments on the Quality of English Language

English language is sufficient

Reviewer 3 Report

Comments and Suggestions for Authors

Thank you for taking my suggestions into account and implementing them in your manuscript. I believe the revisions have improved the overall quality of the work.
